# Comparison of Gene Expression Changes in Three Wheat Varieties with Different Susceptibilities to Heat Stress Using RNA-Seq Analysis

**DOI:** 10.3390/ijms231810734

**Published:** 2022-09-14

**Authors:** Myoung Hui Lee, Kyeong-Min Kim, Wan-Gyu Sang, Chon-Sik Kang, Changhyun Choi

**Affiliations:** National Institute of Crop Science, Rural Development Administration, Wanju-gun 55365, Korea

**Keywords:** heat stress, hexaploid wheat, seedling stage, RNA-sequencing, ROS-scavenging enzyme, heat shock proteins, heat stress transcription factors

## Abstract

Wheat is highly susceptible to heat stress, which significantly reduces grain yield. In this study, we used RNA-seq technology to analyze the transcript expression at three different time-points after heat treatment in three cultivars differing in their susceptibility to heat stress: Jopum, Keumkang, and Olgeuru. A total of 11,751, 8850, and 14,711; 10,959, 7946, and 14,205; and 22,895, 13,060, and 19,408 differentially-expressed genes (log2 fold-change > 1 and FDR (padj) < 0.05) were identified in Jopum, Keumkang, and Olgeuru in the control vs. 6-h, in the control vs. 12-h, and in the 6-h vs. 12-h heat treatment, respectively. Functional enrichment analysis showed that the biological processes for DEGs, such as the cellular response to heat and oxidative stress—and including the removal of superoxide radicals and the positive regulation of superoxide dismutase activity—were significantly enriched among the three comparisons in all three cultivars. Furthermore, we investigated the differential expression patterns of reactive oxygen species (ROS)-scavenging enzymes, heat shock proteins, and heat-stress transcription factors using qRT-PCR to confirm the differences in gene expression among the three varieties under heat stress. This study contributes to a better understanding of the wheat heat-stress response at the early growth stage and the varietal differences in heat tolerance.

## 1. Introduction

Global climate change is expected to have a direct effect on crop production around the world. In particular, wheat is highly susceptible to heat stress, and it is estimated that global wheat production will decrease by 4% to 8.5%/°C increase in temperature [1,2,3]. Among abiotic stress factors, heat is one of the most widely studied in plants. Thus, it is well known that heat causes morphological, physiological, and biochemical changes that lead to significant reductions in wheat yield [4]. The effect of heat on plants depends on the duration of the exposure to heat and the specific plant growth-stage at which the stress is experienced [5,6,7,8,9,10]. The optimum temperature range for wheat flowering and grain filling is 12–22 °C [11]. However, an increase in temperature of 1–2 °C shortens the grain filling period, resulting in lower grain weight [12]. In addition, grain yield can be significantly reduced by short-term exposure to high (>35 °C) temperatures [13]. Overall, heat stress disrupts productivity at all developmental stages, including embryonic cells [14], the early stages of meiosis [15], microspore and pollen cell development at the time of floral initiation [16], grain filling [17], root growth [18], and the survival of productive tillers [19], all of which result in the reduction of grain yield. Additionally, heat causes the interruption of protein folding and synthesis [11], leading to the production and accumulation of stressing agents that immediately disrupt major cellular metabolic processes, as well as DNA replication, transcription, and mRNA transport and translation until the cells recuperate [20]. Furthermore, heat shock causes changes in membrane potential (depolarization), lipid peroxidation, protein oxidation, and damage to nucleic acids. Heat significantly reduces the relative leaf chlorophyll content at all growth stages, and accelerates leaf senescence [21]. Overall, a moderate heat treatment applied for an extended period causes gradual senescence, whereas an extremely high temperature applied for a short period leads to protein denaturation and aggregation, causing rapid plant death [4,22,23]. High temperatures reduce plant water content, leading to cell dehydration [24]. Water movement in and out of the cell occurs via aquaporins, a group of proteins that belong to the highly-conserved membrane protein superfamily known as the major intrinsic proteins, which are recognized as the most abundant transmembrane transporters of substrates such as water, glycerol, urea, CO_2_, NH_3_, reactive oxygen species (ROS), and metalloids [25].

Heat-induced ROS include singlet oxygen (^1^O_2_), superoxide (O_2_•−), and hydroxyl radicals (•OH) [26,27], which need to be neutralized by the cellular antioxidant system to protect plants from oxidative damage [28]; thus, antioxidant enzyme activities, including ascorbate peroxidase (APX), catalase (CAT), glutathione reductase (GR), glutathione peroxidase (GPX), superoxide dismutase (SOD), and peroxidase (POX), are increased by heat stress [7,29]. Specifically, glutathione S-transferase (GST), APX, and CAT activities reportedly increase in heat-tolerant wheat cultivars [30]. To eliminate accumulated ROS and maintain metabolic activity and productivity, plants have a well-organized defense system composed mainly of transcription factors (TFs) and heat shock proteins—HSPs [31,32] that function as molecular chaperones protecting plants from heat stress by maintaining proteins in their functional conformation. A total of 753 *TaHSPs* have been identified in the wheat genome, including 169 *TaSHSP*, 273 *TaHSP40*, 95 *TaHSP60*, 114 *TaHSP70*, 18 *TaHSP90,* and 84 *TaHSP100* [33]. Furthermore, these HSPs show tissue-specific and developmental-stage-specific expression [34], and when heat stress occurs, HSPs are rapidly induced by the transcriptional activity of heat stress transcription factors (HSFs). However, despite being the most important staple food crop globally, very limited information is available regarding the wheat HSP family. Recently, an analysis by a genome-wide association study was performed to identify the gene response to heat stress in wheat during flowering [35] and seed development [33].

In addition to plant genotype, the effects of heat shock depend on heat intensity, timing, and duration; therefore, the development of tolerant varieties should contribute to minimizing the effects of heat stress. Accordingly, Lu et al. [36] reported that wheat genotypes that were heat tolerant at the seedling stage showed higher grain yields after a heat stress treatment, suggesting that early selection for heat-tolerance might be effective in breeding for heat tolerance. However, heat stress effects at the seedling stage have not been well characterized in wheat. In addition, heat-related gene expression in wheat cultivars remains poorly understood. In this study, we analyzed the transcriptome of wheat seedlings under heat stress using RNA-seq technology. Three varieties with varying degrees of susceptibility to heat were subjected to differential gene-expression analysis at two different sampling time-points after heat treatment initiation to identify heat-induced differentially-expressed genes (DEGs) within each genotype. We identified thousands of DEGs, which were evaluated using volcano plots, cluster plots, heatmaps, and GO enrichment analyses. We further analyzed the relative expression levels of genes related to transcription factors, HSPs, antioxidant enzymes, and detoxification. Our results provide new insights into the molecular mechanisms underlying the heat-stress response in wheat at the seedling stage, and the differences in susceptibility to heat among wheat varieties.

## 2. Results

### 2.1. Evaluation of the Heat-Stress Response in the Wheat Varieties Tested

Previously, Ko et al. [37] reported the effects of high temperature on chlorophyll content, grain weight, and shoot dry weight at the early grain filling stage in 11 Korean wheat varieties. Based on the rate of reduction observed in these parameters, cultivars of Jopum (JP), Keumkang (KK), and Olgeuru (OL) were classified as heat tolerant, moderately heat tolerant, and heat susceptible, respectively [37]. ‘OL’ was obtained by crossing ‘Geuru’ (susceptibility) and ‘Sakai 143’ (susceptibility); ‘Keumkang’ was obtained by crossing varieties of ‘Sakai 75’ (resistance) and ‘Geuru’ (susceptibility); and ‘Jopum’ was inherited from ‘Sakai 75’ (resistance) [37]. In this study, we used these varieties and an RNA-seq expression analysis to compare their gene expression profiles in young seedlings growing under heat stress. First, to confirm the heat response of young seedlings of the three varieties, we observed the extent of leaf drying after repeated heat treatment for 3 h at 45 °C and their subsequent recovery three days after return to normal temperature conditions. Differences in the green index among the three varieties were not significant after the first or second heat treatments; however, after the third heat treatment, the green index of OL decreased compared to those of JP and KK, and after the fourth heat treatment, JP showed only a few dry leaves, KK looked normal, and OL showed many dry leaves (Figure 1A); concomitantly, green leaves of untreated plants from the three cultivars became more abundant as plants continued to grow (Figure 1B). Seedlings of OL showed a significant decrease in green index compared with those of JP (Figure 1C). Consistent with the abovementioned classification of the tested varieties in terms of their level of heat tolerance, our results indicate that, with respect to young seedlings—as might be expected—the tested cultivars ranked in the following order in extent of leaf senescence observed in response to heat stress: JP < KK < OL.

### 2.2. Differential Expression and Cluster Analyses of Inducible Genes by RNA-Seq

To identify DEGs in 6-h (H6) and 12-h (H12) heat-treated seedlings of JP, KK, and OL relative to the untreated control seedlings (CT), we calculated the gene transcript abundance using Kallisto (v. 0.46.0) [38] (Appendix A) and identified DEGs in R (v. 3.6.3) by setting a threshold of log2 fold-change > 1 and FDR (padj) < 0.05. Differentially-expressed genes (DEG) were identified by pairwise comparisons of the three experimental conditions in all three cultivar datasets separately. As shown in Figure 2 and Appendix A, relative to the control seedlings, JP showed 11,751 (7022/4729 up-/downregulated), 8850 (3368/5482 up-/downregulated), and 14,711 (5412/9299 up/downregulated) DEGs in CT vs. H6, CT vs. H12, and H6 vs. H12, respectively. In the case of KK, there were 10,959 (6640/4319 up-/downregulated), 8,850 (3621/4325 up-/downregulated), and 14,205 (5728/8477 up-/downregulated) DEGs in CT vs. H6, CT vs. H12, and H6 vs. H12, respectively. Lastly, in OL, we found 22,895 (11,302/11,593 up-/downregulated), 13,060 (6201/6859 up-/downregulated), and 19,408 (9102/10,308 up-/downregulated) DEGs in CT vs. H6, CT vs. H12, and H6 vs. H12, respectively.

To provide further insight into the genes whose expression changed after heat treatment, we used Trinity (version 2.14.0) [39] to group DEGs into five clusters according to the change in their expression pattern (Appendix A). The results showed that genes in cluster 1 showed a decreasing expression at H6 and H12 compared to the CT group. Specifically, genes in cluster 4 showed significant downregulation at 6 h after heat treatment, followed by upregulation thereafter. In contrast, genes in cluster 2 consistently showed upregulated or constant levels of expression. Meanwhile, genes in cluster 3 were upregulated at H6, and then downregulated at H12. Most DEGs identified in OL were grouped in cluster 1 (66.14%), followed by those grouped in cluster 2 (27.18%), with very few DEGs in clusters 3–5 (Appendix A). Conversely, most DEGs identified in JP were grouped in cluster 1, followed by those grouped in cluster 2 (Appendix A). Interestingly, in KK, DEGs were predominantly grouped in cluster 5 (78.73%), and to a lesser proportion in clusters 1 and 2 (Appendix A).

### 2.3. Functional Annotation of the DEGs and Shared and Specific Biological Processes (BPs)

To elucidate the biological functions related to heat stress at the molecular level, genes that were differentially expressed in Jopum, Keumkang, and Olgeuru were included in a functional enrichment analysis performed using the GO databases (Appendix A). GO terms were grouped using REVIGO [40] to remove the redundancy.

In Jopum, 133, 151, and 213 unique biological processes (BPs) were identified in CT vs. H6, CT vs. H12, and H6 vs. H12, respectively; thirty-eight unique BPs were significantly enriched across all comparisons, which included a cellular response to heat (GO:0034605), response to hydrogen peroxide (GO:0042542), and cellular response to oxidative stress (GO:0034599) (Figure 3A–C, Appendix A). The top 20 significantly-enriched BPs under each comparison of Jopum are shown in Figure 3 and Appendix A. DEGs identified in all three comparisons in Jopum were represented as Venn diagrams (Figure 4A, Appendix A). At least two comparisons shared 133 BPs that were common; the number of specifically-enriched BPs under each of the comparisons was 34 in CT vs. H6, 33 in CT vs. H12, and 64 in H6 vs. H12 (Figure 4A).

In Keumkang, 150, 107, and 201 DEGs were identified in CT vs. H6, CT vs. H12, and H6 vs. H12, respectively (Appendix A, Appendix A). Thirty-four unique BPs were significantly-enriched across all comparisons, which included a response to hydrogen peroxide (GO:0042542) and photorespiration (GO:0009853). The top 20 significantly-enriched BPs under each comparison of Keumkang are shown in Figure 3D–F and Appendix A. Furthermore, 45, 30, and 78 BPs were specifically enriched in CT vs. H6, CT vs. H12, and H6 vs. H12, respectively (Figure 4B).

In Olgeuru, 209, 221, and 148 DEGs were identified in comparisons of CT vs. H6, CT vs. H12, and H6 vs. H12, respectively (Figure 4C, Appendix A, Appendix A). The top 20 significantly-enriched BPs in each of the three comparisons are shown in Figure 3G–I. Thirty-nine unique BPs were significantly enriched across all three comparisons, which included a cellular response to oxidative stress (GO: GO:0034599). Seventy-six BPs were identified in at least two comparisons. Furthermore, 50, 97, and 42 BPs were specifically-enriched in CT vs. H6, CT vs. H12, and H6 vs. H12, respectively (Appendix A, Appendix A).

The following 8 BP terms were commonly enriched in Jopum, Keumkang, and Olgeuru in all three comparisons: protein folding (GO:0006457 photosynthesis, light harvesting (GO:0009765), photosynthesis, light harvesting in photosystem I (GO:0009768), photorespiration (GO:0009853), carbon fixation (GO:0015977), photosynthesis (GO:0015979), protein-chromophore linkage (GO:0018298), and protein refolding (GO:0042026). The number of commonly-enriched BPs under all three comparisons was eight BPs between Jopum and Keumkang, eight BPs between Keumkang and Olgeuru, and six BPs between Olgeuru and Jopum (Figure 4D, Appendix A).

A total of 65 and 78 DEGs were shared by Jopum and Keumkang in CT vs. H6 and CT vs. H12, respectively; 68 (~51%) and 85 (~45%) DEGs were differentially expressed between Jopum and Keumkang in CT vs. H6. In addition, 73 (~48%) and 29 (~27%) DEGs were differentially expressed between Jopum and Keumkang in CT vs. H12 (Figure 4E). A total of 84 and 48 DEGs were shared between Keumkang and Olgeuru in CT vs. H6 and CT vs. H12, respectively. In addition, 66 (~44%) and 135 (~38%) DEGs were identified in CT vs. H6, and 59 (~55%) and 173 (~22%) DEGs were identified in CT vs. H12 (Figure 4E). Between Olgeuru and Jopum, a total of 72 and 68 DEGs were shared in CT vs. H6 and CT vs. H12, respectively; 138 (~66%) and 61 (~51%) DEGs were identified in CT vs. H6. In addition, 153 (~69%) and 83 (~55%) DEGs were identified in CT vs. H12 (Figure 4E).

The analysis of significantly-enriched GO functions with respect to ROS metabolic processes (GO:1901031, GO:0000302, GO:0019430, GO:1901671) and cellular response to heat (GO:0009408, GO:0034605) were performed for all of the comparative combinations (Appendix A). These genes were significantly enriched among the three comparisons as well as the three cultivars (Appendix A).

### 2.4. Expression of Antioxidant Enzyme-Related Genes and ROS Detoxification

As per RNA-seq analysis, the major genes encoding enzymes involved in ROS scavenging, catalase (CAT), ascorbate peroxidase (APX), glutathione S-transferase (GST), superoxide dismutase (SOD), glutathione peroxidase (GPX), and peroxidase (POX) were differentially expressed after heat treatment compared to the controls (Figure 5, Appendix A). Nine *CATs* were differentially expressed under high temperatures and, among them, three DEGs, TraesCS5A02G498000.1, TraesCS4D02G322700.1, and TraesCS4B02G325800.1, showed similar patterns of expression in the three cultivars. However, six DEGs, including TraesCSU02G105300.1, TraesCS6D02G048300.1, TraesCS7A02G549800.1, TraesCS7A02G549800.2, TraesCS6D02G048300.1, TraesCS6D02G048300.3, and TraesCS6A02G041700.1, were upregulated in JP but downregulated in OL. In addition, 12 APXs were differentially expressed under heat treatment, although there was no significant difference among the three varieties in expression of three *tAPXs,* namely TraesCS6D02G397200.1, TraesCS6D02G397200.2, and TraesCS6B02G462300.1; three *APX6*, i.e., TraesCS7D02G249200.1. TraesCS7A02G250500.2, and TraesCS7B02G140600.2; and one *APX2,* TraesCS2D02G080000.1. Furthermore, four *APX4* genes (TraesCS2A02G437700.1, TraesCS2D02G434500.1, TraesCS2B02G457600.1, and TraesCS6A02G412900.1) and one *APX2* (TraesCS2B02G096200.1) were downregulated in JP but upregulated in OL. Finally, all eight *POX* genes (TraesCS6D02G303900.1, TraesCS2D02G107600.1, TraesCS1D02G096400.1, TraesCS7D02G347300.1, TraesCS1A02G077700.2, TraesCS2B02G125800.1, TraesCS2D02G108500.1, and TraesCS6B02G063900.1), one of the two *SOD* (TraesCS4D02G242800.1) genes, and one of the five *GPX* (TraesCS6D02G228800.1) genes showed different expression patterns after heat treatment in all three cultivars.

### 2.5. Dynamic Expression of Transcription Factors and Heat Shock Proteins

We identified 19 heat stress transcription factors (HSFs) in DEGs that were analyzed (Figure 6A and Appendix A) after heat treatment. Of the 19 HSFs identified by RNA-seq analysis, 6, 3, and 6 were annotated as rice Hsf sp17, HsfB-2c, and Hsf-2C, respectively. In addition, one HsfA group (*HsfA-2c*, TraesCS1A02G375600), HsfB group (*HsfB2b*, TraesCS7A02G270100.1), and HsfC group (*HsfC-1b*, TraesCS3A02G280800.1) were identified. The relative expression of *HsfB-2b* and *HsfC-1b* was also significantly downregulated by heat stress, but the expression patterns did not differ among varieties.

We also identified 30 HSPs that were highly responsive to heat stress at 6 h from treatment initiation, compared with their basal levels of expression in the three varieties. Interestingly, most expression levels detected by RNA-seq analysis suggested downregulation (Figure 6B, Appendix A). One *HSP101* (TraesCS3D02G273600.1), *HSP90* (TraesCS2B02G047400.1), four *HSP70* (TraesCS4A02G066100.1, TraesCS4B02G243400.1, TraesCS4D02G243000.1, and TraesCS1D02G284000.2), and eight *sHSPs* (TraesCS6D02G169100.1, TraesCS7A02G202200.1, TraesCS3A02G033900.1, TraesCS4B02G225400.1, TraesCS7A02G177700.1, TraesCS3D02G114900.1, TraesCS6D02G322300.1, and TraesCS4D02G226000.1) were significantly decreased after 6 h of heat treatment. Unfortunately, however, corresponding data could not be obtained for these genes because the filtering CPM (counts per million) did not exceed 2 after 12 h of heat treatment. In contrast, four small HSPs (*sHSP;* TraesCS1D02G319400.1, TraesCS1B02G331900.1, TraesCS1A02G319600.1, and TraesCS2B02G247300.1) increased to a similar extent at 6 h in the three varieties but showed a pattern similar to that of the control group after 12 h of heat treatment (Figure 6B). One *HSP90,* TraesCS5B02G258900.1, showed different expression patterns in JP and OL. The expression of this gene increased at 6 h in both JP and OL and continued to increase at 12 h in JP, but decreased in OL (Figure 6B).

### 2.6. Quantitative Real-Time PCR Analysis to Validate Gene Expression Levels Found by RNA-Seq Analysis

To validate the gene expression levels obtained by RNA-seq analysis, 21 DEGs were selected to perform quantitative real-time polymerase chain reaction (qRT-PCR) analysis. Subsequently, we determined the transcriptional levels of ROS-scavenging genes encoding *APX* (TraesCS2B02G457600.1, TraesCS6A02G412900.1), *CAT* (TraesCS6D02G048300, TraesCS7A02G549800, and TraesCS5A02G498000.1), *GPX* (TraesCS2B02G604800 and TraesCS2A02G582100.1), GST (TraesCS3D02G299900.1), *GR* (TraesCS6A02G383800.2), and *POX* (TraesCS1A02G077700.2, TraesCS1D02G096400.1, and TraesCS6B02G303900.1) (Figure 7). Two *CATs* (*CAT6D*: TraesCS6D02G048300 and, *CAT-7A*: TraesCS7A02G549800) and one *POX* (*POX-1D*: TraeCS1D02G096400.1) showed progressive and time-dependent expression increases in JP and KK, but with different expression patterns in OL (Figure 7C,D,K). In all three varieties, the expression of two *GPXs* (*GPX-2A*: TraesCS2A02G582100.1 and *GPX-2B*: TraesCS2B02G604800) was progressively reduced in a time-dependent manner by heat treatment (Figure 7G–H). In turn, the transcription levels of the three *POX* genes were significantly lower in OL than in JP or KK (Figure 7J–L). Meanwhile, the expression of one *POX* (*POX-6B*: TraeCS6B02G303900.1) decreased progressively in both JP and KK, but showed very low basal expression levels in OL, and was observed showing a slight increase at 3 h and 6 h, and then a decrease in expression level (Figure 7L). Lastly, the expression of other genes showed slightly different patterns depending on the variety, but the differences were not significant. Specifically, the transcriptional levels of *CAT-5A* and *POX-1D* in OL were not detected after 9 h of heat treatment (Figure 7E,K).

Transcript levels of *HSP* and *HSF* genes, which are up- or downregulated upon heat treatment, were also investigated by DEG analysis. We selected six highly-responsive genes to heat stress, as well as three previously reported genes: *TaHSF3* [41], *TaHsfA6f* [42], and *TaHSP23.9* [43] (Figure 8). The transcript levels of five *HSPs*, *HSP101c*, *HSP90.1-B1*, *HSP90.6*, *HSP70-8*, and *TaHSP23.9*, were markedly increased 3 h after heat treatment, and then gradually decreased in all three cultivars (Figure 8E–I). Similarly, the transcript levels of *HSFA-2c* increased in the three varieties after 3 h of heat treatment, and then decreased in KK and OL, but not in JP (Figure 8A). Similarly, the relative expression levels of *HsfB-2b* and *TaHSF3* significantly increased after 3 h of heat treatment in KK and OL and then decreased, but increased up to 6 h in JP and decreased thereafter; furthermore, their expression levels were lower than those in JP and OL (Figure 8B,C). The transcript level of *TaHsfA6f* was higher under basal conditions and then gradually decreased with increasing duration of heat treatment (Figure 8D). These results indicate that the expression of *HSFs* and *HSP* genes varied depending on the variety after heat treatment, with KK showing the highest level of expression for these genes, followed by OL, and JP was the lowest.

Transcript levels of *HSP* and *HSF* genes, which are up- or downregulated upon heat treatment, were also investigated by DEG analysis. We selected six highly-responsive genes to heat stress, as well as three previously reported genes: *TaHSF3* [41], *TaHsfA6f* [42], and *TaHSP23.9* [43] (Figure 8). The transcript levels of five *HSPs*, *HSP101c*, *HSP90.1-B1*, *HSP90.6*, *HSP70-8*, and *TaHSP23.9*, were markedly increased 3 h after heat treatment, and then gradually decreased in all three cultivars (Figure 8E–I). Similarly, the transcript levels of *HSFA-2c* increased in the three varieties after 3 h of heat treatment, and then decreased in KK and OL, but not in JP (Figure 8A). Similarly, the relative expression levels of *HsfB-2b* and *TaHSF3* significantly increased after 3 h of heat treatment in KK and OL and then decreased, but increased up to 6 h in JP and decreased thereafter; furthermore, their expression levels were lower than those in JP and OL (Figure 8B,C). The transcript level of *TaHsfA6f* was higher under basal conditions and then gradually decreased with increasing duration of heat treatment (Figure 8D). These results indicate that the expression of *HSFs* and *HSP* genes varied depending on the variety after heat treatment, with KK showing the highest level of expression for these genes, followed by OL, and JP was the lowest.

Transcript levels of *HSP* and *HSF* genes, which are up- or downregulated upon heat treatment, were also investigated by DEG analysis. We selected six highly-responsive genes to heat stress, as well as three previously reported genes: *TaHSF3* [41], *TaHsfA6f* [42], and *TaHSP23.9* [43] (Figure 8). The transcript levels of five *HSPs*, *HSP101c*, *HSP90.1-B1*, *HSP90.6*, *HSP70-8*, and *TaHSP23.9*, were markedly increased 3 h after heat treatment, and then gradually decreased in all three cultivars (Figure 8E–I). Similarly, the transcript levels of *HSFA-2c* increased in the three varieties after 3 h of heat treatment, and then decreased in KK and OL, but not in JP (Figure 8A). Similarly, the relative expression levels of *HsfB-2b* and *TaHSF3* significantly increased after 3 h of heat treatment in KK and OL and then decreased, but increased up to 6 h in JP and decreased thereafter; furthermore, their expression levels were lower than those in JP and OL (Figure 8B,C). The transcript level of *TaHsfA6f* was higher under basal conditions and then gradually decreased with increasing duration of heat treatment (Figure 8D). These results indicate that the expression of *HSFs* and *HSP* genes varied depending on the variety after heat treatment, with KK showing the highest level of expression for these genes, followed by OL, and JP was the lowest.

## 3. Discussion

Heat stress is generally considered the most detrimental abiotic stress factor for economically important crops, especially wheat [44]. Padaria et al. [45] compared the differences in expression profiles induced by heat treatment at different developmental stages. However, related transcriptional profiles in wheat have not been widely studied among cultivars, particularly for comparison purposes. RNA sequencing (RNA-seq) has been primarily used to identify novel and conserved stress-responsive genes associated with heat stress tolerance in wheat [33,35,45,46]. In this study, we evaluated heat-stress-induced gene expression by RNA sequencing over time after heat treatment in three cultivars differing in susceptibility to heat stress. The mapping ratio of the sample to the reference genome ranged from ~50% to 86%, with an average of 80% (Appendix A). To further identify DEGs at 6 h (H6) and 12 h (H12) after heat-treatment initiation relative to the untreated control (CT) seedlings, we analyzed DEGs using a volcano plot (Figure 2). There were significant differences in the number of DEGs in the H6 vs. CT, H12 vs. CT, and H6 vs. H12 groups. These results indicate the differences in wheat regulatory mechanisms in response to heat under different treatment durations. The analysis of significantly-enriched GO functions with respect to ROS metabolic processes (GO:1901031, GO:0000302, GO:0019430, GO:1901671) and cellular response to heat (GO:0009408, GO:0034605) was significantly enriched among the three comparisons and three cultivars. Thus, we identified the DEGs associated with enzymatic ROS-scavenging activity, HSPs, and HSFs that respond significantly to heat stress.

A direct result of stress-induced cellular changes is the overproduction of ROS in plant tissues, which are continuously reduced/eliminated by the plant antioxidant system, thus maintaining a steady-state level of ROS under stress conditions [47]. Consistently, an increased activity of many antioxidant enzymes has been observed in many plants [48,49]. Here, RNA-seq analysis showed that nine *CATs*, twelve *APXs*, seven *POX*, two *SODs*, five *GPXs*, two *GSTs,* and one *GR* were identified in the three varieties under study, which were highly responsive to heat treatment. Furthermore, these DEGs were validated by qRT-PCR. Transcript levels of the identified DEGs showed various expression patterns in the three cultivars. In particular, two *CAT* (*CAT-6D*:TraeCS6D02G048300 and *CAT-7A*: TraeCS7A02G549800) and one *POX* (*POX-1D*: TraeCS1D02G096400.1) were upregulated by heat treatment, with their expression levels differing by cultivar (Figure 7).

Previously, Schleiff et al. [50] reported that HSFs are primarily involved in stimulating the rapid synthesis and accumulation of HSPs. Indeed, HSFs are rapidly induced within 1 h of heat stress initiation to activate downstream pathways [44]. Thus, 56 *HSF* genes have been identified in wheat [51] and divided into 3 classes: *HsfA, HsfB*, and *HsfC*. However, few *Hsfs* have been cloned and characterized because of the complexity of the wheat genome. Particularly, *TaHsfA-2d* [52], *TaHsfA6b* [53], *TaHsfA2-1* [54], *TaHsfA2e-5D* [55], and *TaHSFA6f* [43] reportedly play important roles in heat stress. Using RNA-seq analysis, here, we identified three new candidate genes that might be involved in the heat-stress response of wheat young seedlings. One from the *HsfA* group (*HsfA-6e*, TraesCS1A02G375600), one from the *HsfB* group (*HsfB-2b*, TraesCS7A02G270100.1), and one from the *HsfC* group (*HsfC-1b*, TraesCS3A02G280800.1). As per qRT-PCR analysis, the relative expression of *HsfB-2b* and *HsfC-1b* was significantly increased upon heat stress, but the expression pattern differed by cultivar. The *HSF* genes identified and investigated in this study suggest that they play positive roles in regulating thermotolerance, especially in wheat. Plants synthesize many stress-responsive proteins, including HSPs, in response to heat stress. These HSPs exhibit tissue-specific and developmental-stage-specific expression [34]. However, in wheat, very limited information is available regarding the HSP family, although it is by and large the most important staple food crop in the world. Here, using RNA-seq analysis, we identified a new candidate gene involved in the heat-stress response in wheat plants and evaluated it by qRT-PCR analysis.

As a result of RNA-seq and qRT-PCR analyses, we showed that the expression level of *HSF* and *HSP* genes after heat treatment differed by variety. However, the ranking of varieties for gene expression level and phenotype did not match the degree of heat sensitivity estimated. Nonetheless, these results will help understand the role of HSPs in wheat under heat stress conditions. Moreover, our findings show that, although RNA-seq methods and data analysis approaches allowed the identification of a significant number of expressed genes, there were cases in which the results observed were not consistent with the results obtained by qRT-PCR analysis, suggesting that further study is required.

## 4. Materials and Methods

### 4.1. Evaluation of Heat Stress in Three Wheat Varieties

To investigate leaf drying after heat treatment, a 50-hold plastic tray (54 × 28 × 6 cm) was filled with a 1:1 substrate garden bed soil (Punong, Kyoungju, Korea) and agricultural soil (Seoul Bio, Korea), and the seeds of three Korean wheat varieties [56], ‘Jopum’ (JP, accession no. 2002-874), ‘Keumkang’ (KK, accession no. 1997-4435), and ‘Olgeuru’ (OL, accession no. 1997-4432), were sown in a row; ten seeds were shown per variety. To ensure uniform germination, the trays with the sown seeds were incubated at 4 °C for three weeks in a plant growth chamber (Dasol Scientific, Hwaseong, Korea), and for a further two days in a controlled growth room at 22°C under a 22-h photoperiod [57]. Trays were watered from underneath and maintained under well-watered conditions. Heat treatment was performed at 45 °C for 3 h in a growth chamber (Dasol Scientific, Korea), followed by recovery for three days in the controlled growth room. The treatment was repeated for three additional cycles of heat treatment and recovery periods, respectively, with the same set of seedlings. The experiment was repeated three times. The greenness index was calculated as a percentage by dividing the area of the green color using the formula proposed by Laohaichi et al. [58] by different numbers of the plant RGB image+ and openCV of Python 3.8 [59].

### 4.2. RNA Preparation and Quality Control for RNA-Seq Transcriptional Profiling

For RNA extraction, seedlings of the three cultivars, JP, KK, and OL, were grown in pots for 19 days at 22 °C under a 22-h photoperiod in the controlled growth room [57]. The seedlings were subjected to heat treatment at 35 °C [60] for 6 and 12 h; seedlings without heat treatment served as the control. Subsequently, leaves from the control and heat-treated plants were collected and rapidly frozen in liquid nitrogen and then stored in a −80 °C freezer until the RNA extraction procedure. Total RNA was extracted using TRIzol^TM^ reagent (Thermo Scientific, Waltham, MA, USA) and the RNA quality and quantity were analyzed using a NanoDrop 1000 spectrophotometer (Thermo Scientific, MA, USA). RNA with an RNA integrity number (RIN) >7.0 was determined using a 2100 Expert Bioanalyzer (Agilent Technologies Inc., Santa Clara, CA, USA) for further use in transcriptome profiling.

### 4.3. cDNA Library Preparation and Illumina Sequencing

A total of 27 RNA-Seq samples from three wheat cultivars were analyzed. A quantity of 1 µg of total RNA was used to construct sequencing libraries using a Truseq Stranded mRNA library prep kit (Illumina, San Diego, CA, USA). Paired-end (2 × 100 bp) reads of samples were sequenced using the Illumina NovaSeq 6000 platform. Low-quality reads, redundant reads, and adapter sequences were eliminated using Trimmomatic (v. 0.38) by following default parameters: removing a read with an average base quality (Q20) below 20 and removing a read less than 50 bp [61]. FastQ files were generated via Illumina bcl2fastq2 (version 2.20) and the quality of trimmed reads was assessed using FastQC (version 0.11.9). To further analyze RNA-Seq data, clean reads were mapped to *Triticum aestivum* genome sequences [62] using Kallisto (version 0.46.0) [38]. Library construction and sequencing were performed by DNA LINK, Inc. (Seoul, Korea).

### 4.4. Differentially-Expressed Gene (DEG) Identification

Transcript abundance was determined using Kallisto (version 0.46.0). A differential gene expression analysis among libraries was performed in R (version 3.6.3) software using the edgeR package (version 3.30) [63]. Differentially-expressed genes (DEG) were identified by pairwise comparisons of the three experimental conditions from the Jopum, Keumkang, and Olgeuru datasets separately. The trimmed means from the M-value (TMM) normalization methods were used to calculate the normalization factors. Genes were filtered by above/below 2-fold, above CPM (counts per million) 2, and FDR (false-discovery rate) < 0.05. The Benjamini–Hochberg method was used to estimate the FDR when identifying DEGs.

### 4.5. Cluster Analysis, GO Enrichment Analysis of DEGs, and Heatmap

A DEG cluster analysis was used to isolate the cluster analysis expression patterns of genes at two different sampling time-points after heat treatment of the three cultivars using Trinity (version 2.14.0 [39]. Hierarchical clustering using the Euclidean method of normalized gene expression was achieved using centralized and log2 (FPKM+1) transforms and tree cutting at 60% depth. Genes were filtered by two conditions of above/below 2-fold change and FDR < 0.05, and grouped under similar expression patterns. For ontology analysis, the Databaseor Annotation, Visualization, and Integrated Discovery (DAVID) v2022q2 [64] was used to select genes with above 2-fold change and FDR (padj) < 0.05. The genes were categorized according to their assigned ontology (GO) terms (cellular components, biological processes, and molecular functions). GO terms were grouped using REVIGO [40] to remove the redundancy. The heatmap was created in MS Excel using conditional formatting and a color scale using selected genes (Appendix A).

### 4.6. Validation of Gene Expression Profile by qRT-PCR

For the validation of gene expression profiles using qRT-PCR analysis, RNA was extracted from 19-day old seedlings (maintained at 22 °C under a 22-h photoperiod in controlled growth room [53]) from the three cultivars that were heat treated at 35 °C for 3, 6, and 9 h in the growth chamber (Dasol Scientific, Korea); subsequently, leaves (Zadoks (Z) growth stage: L4 in Z1.4 and Z2.2 stage) of heat-treated seedlings were collected and rapidly frozen in liquid nitrogen and then stored in a −80 °C freezer until the RNA extraction procedure. Total RNA was extracted using an RNAqueous Total RNA isolation kit (Thermo Scientific, MA, USA) and a TURBO DNA-free kit (Thermo Scientific, MA, USA). RNA quality in the range of 1.8–2.0 at A260/A280 nm and RNA concentrations greater than 100 ng/μL were used. Total RNA (2 μg) was used to prepare cDNA using a high-capacity cDNA reverse transcription kit (Thermo Scientific, MA, USA). To investigate the transcript levels of *HSPs*, ROS-scavenging enzymes, and aquaporins in the leaf tissue, primers were designed using the real-time PCR tool OligoAnalyzer 3.1 (Integrated DNA Technologies, Coralville, IA, USA). qRT-PCR was performed under the following conditions: Initial denaturation at 95 °C for 15 min, 40 cycles of 94 °C for 20 s, 58 °C for 20 s, and 72 °C for 20 s using a SYBR Green Kit (BioFACT, Daejeon, Korea) to detect the transcript levels of each gene on a CFX96 Real-Time PCR Detection System (Bio-Rad, Hercules, CA, USA). The PCR mixture (20 μL) comprised a 50 ng template, 2 pmoles each of forward and reverse primers, and 1× SYBR master mix. To confirm specific amplifications, melting curves were generated by increasing the temperature by 0.5 °C every 5 s from 60 °C to 95 °C. The ΔΔCt method was used as the relative quantification strategy for qRT-PCR data analysis. The primers used for PCR are listed in Appendix A. An ADP-ribosylation factor was used as the internal control [65].

### 4.7. Statistical Analysis

Statistical analyses were performed using the data from experiments performed in triplicate. Data were analyzed using one-way analysis of variance followed by Tukey’s multiple comparison test using PRISM Software 8.0 (GraphPad Software, San Diego, CA, USA). Differences were considered significant at *p* < 0.05.

## Figures and Tables

**Figure 1 ijms-23-10734-f001:**
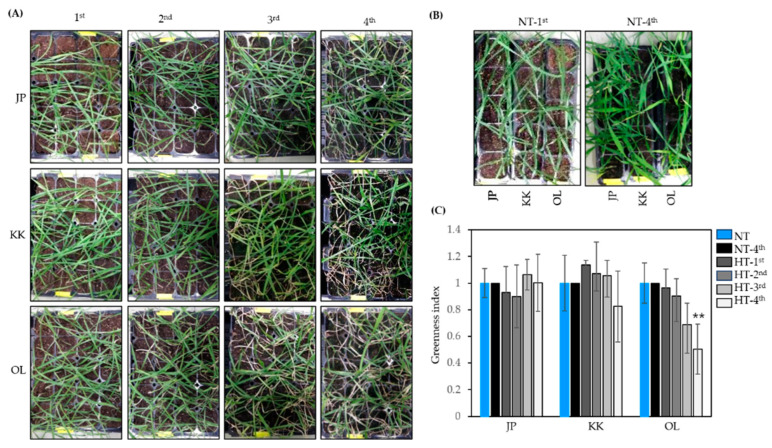
Evaluation of heat stress in three wheat varieties. (**A**) Plant growth after repeated heat treatment at 45 °C for 3 h; (**B**) plant growth at 22 °C; (**C**) greenness index after heat treatment on seedlings of Jopum, Keumkang, and Olgeuru. NT, no-heat treatment; HT, Heat treatment; JP, Jopum; KK, Keumkang; OL, Olgeuru. The 1st, 2nd, 3rd, and 4th indicate repeating 1st, 2nd, 3rd, and 4th heat treatments at 45 °C for 3 h. Asterisks indicate significant differences at *p* < 0.01 (**) as calculated using Student’s *t*-test.

**Figure 2 ijms-23-10734-f002:**
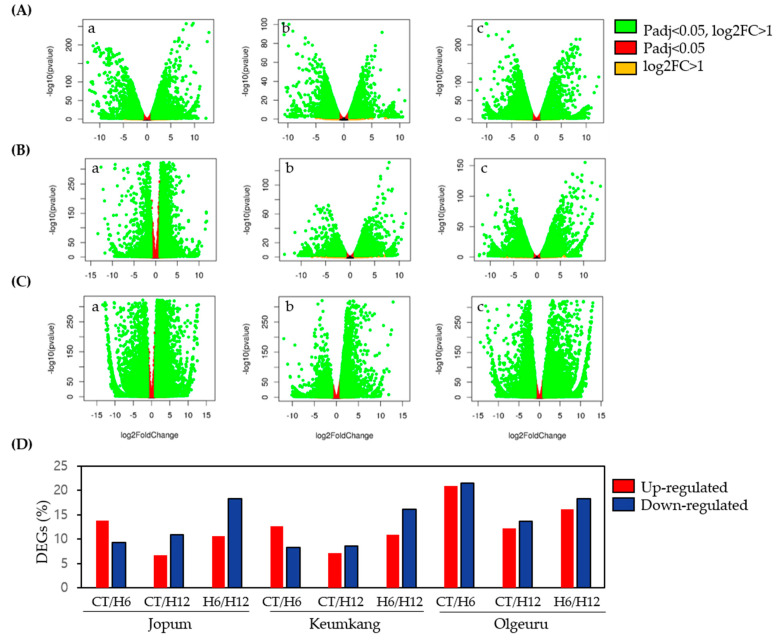
Volcano plot of differentially-expressed genes under heat stress in three cultivars. (**A**–**C**) DEGs in heat-treated groups in Jopum (**A**), Keumkang (**B**), and Olgeuru (**C**). (**D**) The numbers of DEGs found in CT vs. H6 (**a**), CT vs. H12 (**b**), and H6 vs. H12 (**c**) of heat treatment. Screening criterion for DEGs was Padj < 0.05. Log2 ratio stands for log fold changes (FC) using the ratio base 2 logarithm. Padj, adjusted *p*-values; CT, untreated control seedling; H6, 6 h heat treatment; H12, 12 h heat treatment.

**Figure 3 ijms-23-10734-f003:**
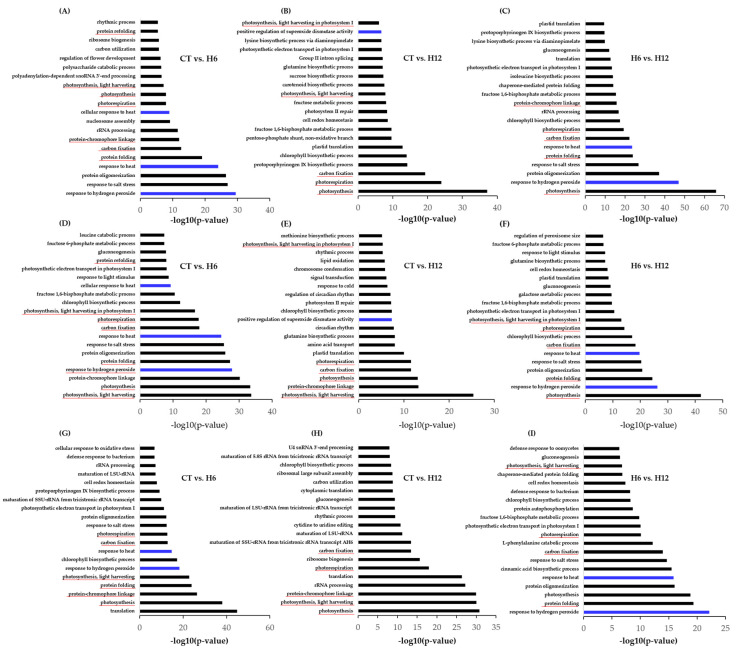
Significantly (FDR (padj) < 0.05)-enriched biological processes (BPs) for differentially-expressed genes (DEGs) of all three comparisons in the three cultivars subjected to heat treatment. (**A**–**C**) The top 20 significantly-enriched BPs for DEGs in Jopum for CT vs. H6, CT vs. H12, and H6 vs. H12, respectively. (**D**–**F**) The top 20 significantly-enriched BPs for DEGs in Keumkang for CT vs. H6, CT vs. H12, and H6 vs. H12, respectively. (**G**–**I**) The top 20 significantly-enriched BPs for DEGs in Olgeuru for CT vs. H6, CT vs. H12, and H6 vs. H12, respectively. BPs underlined in red indicate that they are shared among the three cultivars. Blue bars indicate significantly-enriched BP ‘Cellular response to heat’ or ‘Reactive oxygen species metabolic process’. CT, untreated control; H6, 6-h heat treatment; H12, 12-h heat treatment. Supporting data can be found in Appendix A.

**Figure 4 ijms-23-10734-f004:**
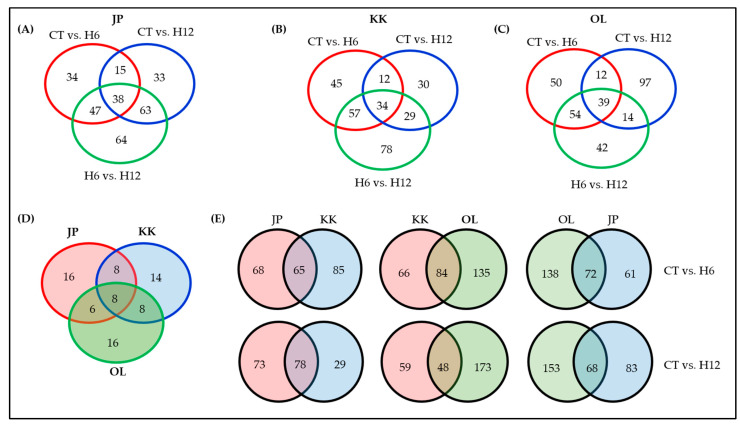
Venn diagram analysis of shared and specific genes across treatments and cultivars. (**A**–**C**) differentially-expressed genes (DEGs) of CT vs. H6, CT vs. H12, and H6 vs. H12 in Jopum (**A**), Keumkang (**B**), and Olgeuru (**C**). (**D**) DEGs of the three comparisons in Jopum, Keumkang, and Olgeuru. (**E**) DEGs between two combinations of JP, KK, and OL cultivars. CT, untreated control; H6, 6 h heat treatment; H12, 12 h heat treatment; JP, Jopum; KK, Keumkang; OL, Olgeuru.

**Figure 5 ijms-23-10734-f005:**
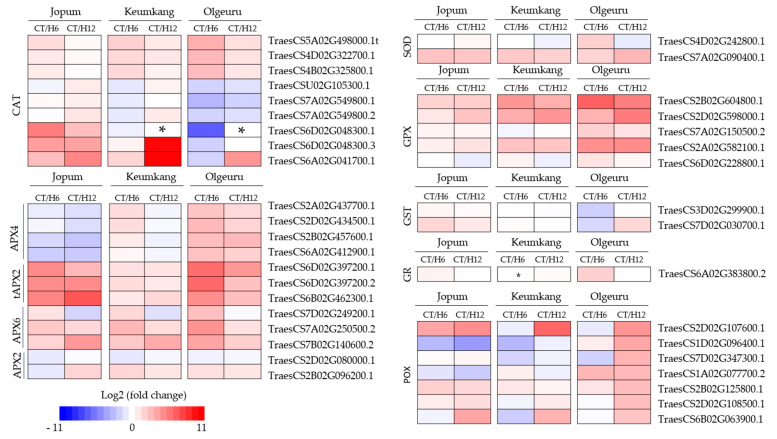
Heatmap of the temporal expression pattern of selected genes involved in ROS scavenging in response to heat stress. Upregulation is indicated in red; downregulation is indicated in blue; *, not evaluated. The scale of the color intensity is presented in the legend. CAT, catalase; APX, ascorbate peroxidase; SOD, superoxide dismutase; GPX, glutathione peroxidase; GST, glutathione S-transferase; GR, glutathione reductase; POX, peroxidase; CT, untreated control; H6, 6 h heat treatment; H12, 12 h heat treatment.

**Figure 6 ijms-23-10734-f006:**
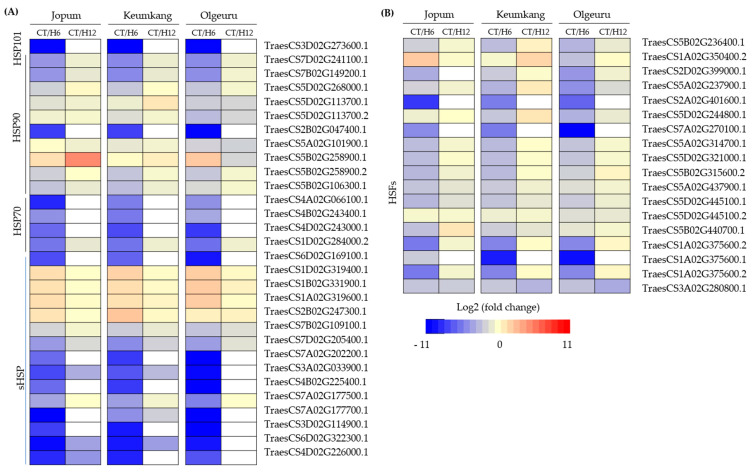
Heatmap of the temporal expression pattern of selected genes encoding HSPs (**A**), and HSFs (**B**). Upregulation is indicated in red; downregulation is indicated in blue; the white box indicates non-validation. The scale of color intensity is shown in the legend. HSP, heat shock protein; sHSP, small heat shock protein; HSFs, heat response transcription factors; CT, untreated control; H6, 6 h heat treatment; H12, 12 h heat treatment.

**Figure 7 ijms-23-10734-f007:**
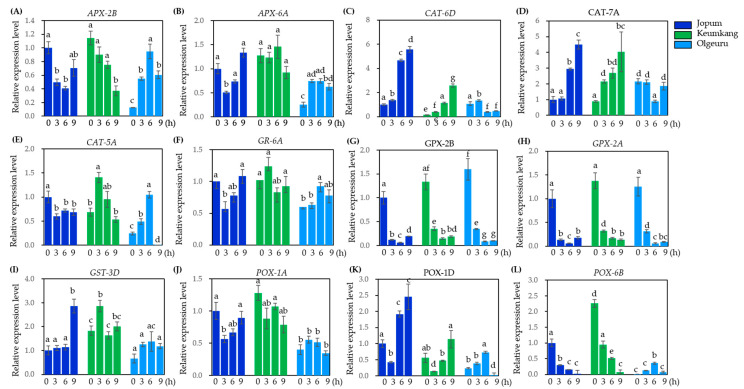
Relative expression level of selected ROS-scavenging genes by RNA-seq analysis. Relative gene expression level of *APXs*, TraesCS2B02G457600.1 (**A**), and TraesCS6A02G412900.1 (**B**); *CATs* TraesCS6D02G0483600 (**C**); TraesCS7A02G549800 (**D**); and TraesCS5A02G49800.1 (**E**); *GR*, TraesCS6A02G383800.2 (**F**); *GPXs* TraesCS5A02G604800 (**G**); and TraesCS2A02G582100.1 (**H**); *GST* TraesCS3D02G299900.1 (**I**); *POXs* TraesCS1A02G77700.2 (**J**); TraesCS1D02G096400.1 (**K**); and TraesCS6B02G303900.1 (**L**). Wheat leaves were heat-treated at 35 °C during 3, 6, and 9 h. Data are representative results from three independent experiments. Error bars represent SD (*n* = 3). Different letters on the bars indicate significant differences between treatments (ANOVA followed by Tukey’s test, *p* < 0.05). APX, ascorbate peroxidase; CAT, catalase; GPX, glutathione peroxidase; GST, glutathione S-transferase; GR, glutathione reductase.

**Figure 8 ijms-23-10734-f008:**
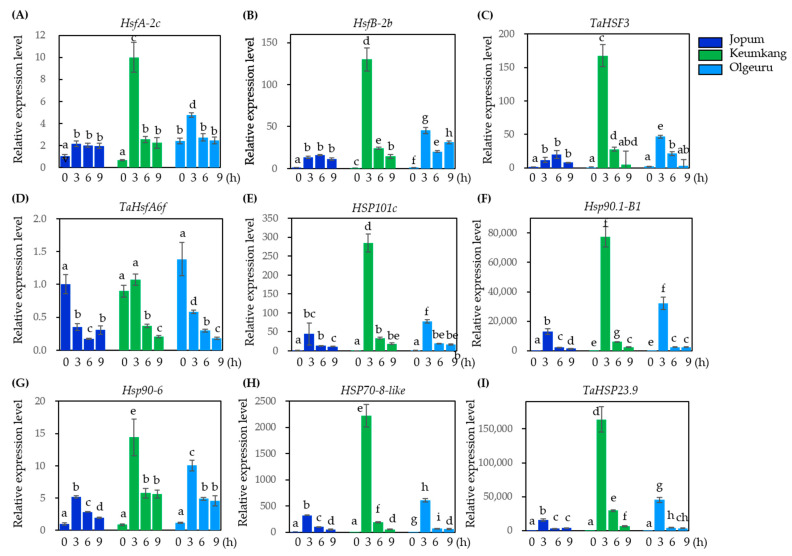
qRT-PCR expression analysis of selected heat shock protein (HSP) and heat-stress transcription factor (HSF) genes in response heat stress in three varieties in RNA-seq data. Relative expression level of *HSFs* TraesCS1A02G375600 (HsfA-2c) (**A**), TraesCS7A02G270100.1 (*HsfB-2b*) (**B**), *TaHSF3* (**C**), *TaHsfA6f* (**D**), *HSFs HSP101* TraesCS3D02G273600.1 (**E**), *HSP90.1B1* TraeCS2B02G474001.1 (**F**), *HSP90.6,* TraesCS5D02G113700.1 (**G**), *HSP70-8* TraesCS4A02G066100.1 (**H**), and *TaHSP23.9* (**I**) in wheat leaves after heat treatment at 35 °C during 3, 6, and 9 h. Data are representative of results from three independent experiments. Error bars represent SD (*n* = 3). Different letters on the bars indicate significant differences between treatments (ANOVA followed by Tukey’s test, *p* < 0.05). HSP, heat shock protein; HSF, heat stress transcription factor.

## Data Availability

Not applicable.

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
