# Peer review of "Comparison of Gene Expression Changes in Three Wheat Varieties with Different Susceptibilities to Heat Stress Using RNA-Seq Analysis"

_ijms, 2022, doi:10.3390/ijms231810734_

Round 1
Reviewer 1 Report (Previous Reviewer 2)
Authors proposed a revised version of their paper.
All the issues were addressed properly.
More explanations where added in the results section.
I have no other issues, therefore the paper deserves to be published.
Author Response
Authors proposed a revised version of their paper.
All the issues were addressed properly.
More explanations where added in the results section.
I have no other issues, therefore the paper deserves to be published.
: Thank you very much for your comments. All additions are marked in blue text.
Reviewer 2 Report (New Reviewer)
1. In the introduction, lines 80-81, I think this sentence should be supplemented with the findings of the study the authors found.
2. In figure 1, the authors should be had a note to describe what is 1st, 2nd, 3rd, and 4th.
3. In results, I think ‘heat resistant’ should be modified to ‘heat tolerant'.
4. In results, the wheat variety Jopum was inherited from the sensibility variety Sakai75, why Jopum is heat resistant?
5. I recommend the authors should improve the resolution of the images in figure 3.
6. In the results, lines 281-300, the authors had written many gene names and codes. I think the authors can use different colors in gene codes to distinguish them in Figure 5.
7. I am curious about whether authors can use data from RNA seq and real-time PCR to clarify why Jopum is high heat tolerance.
Author Response
- In the introduction, lines 80-81, I think this sentence should be supplemented with the findings of the study the authors found.
: This sentence has been moved to the discussion section.
- In figure 1, the authors should be had a note to describe what is 1st, 2nd, 3rd, and 4th.
: The following has been added to the figure legend. "The 1st, 2nd, 3rd, and 4th indicate repeating 1st, 2nd, 3rd, and 4th heat treatment at 45℃ for 3 hours."
- In results, I think ‘heat resistant’ should be modified to ‘heat tolerant'.
: Thanks for the advice. 'Heat resistant' has been changed to 'Heat tolerant'.
- In results, the wheat variety Jopum was inherited from the sensibility variety Sakai75, why Jopum is heat resistant?
: Thanks for pointing out. Saikai 75 is a resistant variety. We have edited 'sensibility' to 'resistance' in line 100.
- I recommend the authors should improve the resolution of the images in figure 3.
: We changed the font to bold and increased the font size. There is not enough space to increase the font size further.
- In the results, lines 281-300, the authors had written many gene names and codes. I think the authors can use different colors in gene codes to distinguish them in Figure 5.
:Corresponding genes are marked in blue and clarifications have been added to figure legend 5.
- I am curious about whether authors can use data from RNA seq and real-time PCR to clarify why Jopum is high heat tolerance.
:Expression of several ROS scavenging genes, HSP, and HSF genes was compared in three wheat varieties with different susceptibility to heat stress. However, it is difficult to conclude that the heat resistance of wheat varieties is determined by these genes. However, we believe that genes differentially expressed to heat stress in the three cultivars have potential as markers of heat tolerance. We expect that RNA-seq could be more accurately differentiated between cultivars rather than chronologically within one cultivar.

Reviewer 3 Report (New Reviewer)
Authors investigated the transcript expression of three wheat cultivars after heat treatment. This study will provide further insights into the wheat heat-stress response. However, there are several major issues need to be solved.
Major
1. In the GO term analysis, for a better interpretation, the GO terms shall be grouped to remove the redundancy using methods such as REVIGO.
2. The data shall be provided into a database such as SRA so that the results can be validated.
Minor
It is unclear which method is used to estimate the FDR in identifying the DEG.
Author Response
Major
- In the GO term analysis, for a better interpretation, the GO terms shall be grouped to remove the redundancy using methods such as REVIGO.
: GO terms were grouped and redundant removed using the REVIGO program (http://revigo.irb.hr). However, we tried to minimize the removal of non-duplicate GO terms. Additional file 4, Additional file 5, Table S3, Table S4, Figure 3, and Figure 4 has been revised. In addition, results section 2.3 has been revised. Additional data and figures have been corrected without tracking function.
- The data shall be provided into a database such as SRA so that the results can be validated.
: We know that raw data is essential to scientific publications for validation in RNA-seq analysis. Unfortunately, we could not keep the raw data after analysis. We tried to upload all the data we had, except the raw data. Please consider these efforts.
Minor
Unfortunately, we could not keep the raw data after analysis. We tried to upload all the data we had, except the raw data. Please consider these efforts.
Minor
It is unclear which method is used to estimate the FDR in identifying the DEG.
: The methods used to estimate the FDR when identifying DEGs were mentioned in Results section 2.2 and Materials and Methods section 4.4.
- Results section 2.2: “identified DEGs in R (v. 3.6.3) by setting a threshold of log2 fold-change > 1 and FDR (padj) < 0.05.”
- Materials and Methods section 4.4: Differential gene expression analysis among libraries was performed in R (version 3.6.3) software (https://cran.r-project.org/src/base/R-3) using the edgeR package (version 3.30) [59]. Genes were filtered by above/ below 2 fold, above CPM (counts per million) 2, and FDR (false-discovery rate) < 0.05.

Round 2
Reviewer 3 Report (New Reviewer)
Thanks for author's effort to improve the manuscript. Some minor issue shall be solved.
1. Authors didn't correctly understand the comment in review round 1:
"It is unclear which method is used to estimate the FDR in identifying the DEG. "
Which method is used for FDR estimation? BH method? It shall be clearly specified.
2. Figure 3 is not correctly displayed. Two figures overlapped are shown in Figure 3.
Author Response
- Authors didn't correctly understand the comment in review round 1:
"It is unclear which method is used to estimate the FDR in identifying the DEG. "
Which method is used for FDR estimation? BH method? It shall be clearly specified.
: Thanks for the kind explanation. As you mentioned, the Benjamini-Hochberg method was used to estimate the FDR when identifying DEGs. Added description in Materials and Methods section 4.4.
- Figure 3 is not correctly displayed. Two figures overlapped are shown in Figure 3.
: Thanks for the advice. Figure 3 has been corrected.

This manuscript is a resubmission of an earlier submission. The following is a list of the peer review reports and author responses from that submission.
Round 1
Reviewer 1 Report
The manuscript “Comparison of gene expression changes in three wheat varieties with different susceptibility to heat stress using RNA-Seq analysis” targets the important question of transcriptional changes in response to heat stress in wheat. To this aim three wheat varieties are studied, which have previously been shown to differ in resilience and response to heat stress. Greenness index and gene expression was compared after subsequent rounds of heat treatments as compared to non- stressed conditions. Further investigations were focused on genes previously described to play roles in heat stress response. Their expression was further analysed using qPCR.
The topic of the manuscript presented is relevant. However, the analysis of data and the presentation would greatly benefit from thorough revision. Major points and concerns are that the novelty of the results is limited, as authors focused on genes previously described to be responsive to heat stress. Based on the data obtained, in depth analysis would have allowed to gain more comprehensive insights into the changes after heat stress, for example by comparing the sets of differentially expressed genes (DEGs) between the different varieties or by investigation of enrichment of GOs or similar based on the DEGs. Most importantly however, methods used in this study are not presented in sufficient detail. As a minor point, the manuscript would benefit from a careful check for typos etc. For detailed comments please see below.
Major comments
1) The data analysis is almost exclusively targeted on groups of genes that were expected to be of interest based on previous knowledge. It would have been interesting to elaborate on the analysis of GO enrichment or protein families or KEGG pathways of DEGs in this context. Furthermore, it would have been of interest to see which DEGs are shared between heat resilient varieties JP and KK and not with OL or if the majority of DEGs are shared.
2) Figure 2: How are non DEGs defined? Are they the genes expressed but not differentially regulated in after stress treatment as compared to the control? Please include an explanation on this.
3) The presentation of genes, gene models, isoforms and splicing variants is partially confusing. It would be beneficial to provide background information (if possible) on how the genomic background varies between the three varieties. This is relevant for estimation of the preciseness of mapping but also for confirmatory qPCR studies.
- line 179: Do TraesCS7A02G549800.1 and TraesCS7A02G549800.2 represent different gene models or different splice variants? How precise is the mapping to the distinct variants? Please clarify.
- line 203f: How was differential splicing of TraesCS1A02G375600.1 addressed? Please clarify.
4) Methods description is largely lacking sufficient information to follow what the authors have done.
- line 402: the Illumina NovaSeq 6000 platform is mentioned, but not the sequencing protocol (e.g. read length, single end, paired end), details on potential multiplexing and sequencing adaptors, etc. Please provide detailed informations.
- line 404: mapping parameters and settings should be provided.
- line 407: EdgeR comprised a variety of different models and methods, please provide all relevant details on the statistics applied, e.g. was it a pairwise comparison, was filtering of low expressed genes applied?
- line 408: what was done with the assembler Trinity and how? In other contexts an assembled transcriptome is not mentioned. Please clarify.
- How was counting of mapped reads performed?
- For sequencing and also for qPCR please indicate quality and quantity of RNA used.
- Please upload raw data to a suitable repository and provide access informations.
- How have heatmaps been done?
- Figure 2: padj implies that adjusted p-values have been used. Please provide details.
- line 149: Please provide sufficient details on how cluster analysis was performed.
- line 219: It is not fully clear what kind of filtering values are meant here? were they removed in filtering of low expressed genes? Or were no significant differences detected? Please clarify.
- discussion: For the qPCR results not showing significant differences between stress/no stress the data should not be discussed as confirmation or validation.
Minor comments:
- line 13ff: context is confusing and should be rephrased. It seems that the numbers provided are average mapping percentages in unstressed control, 6h group, and 12 h group?
- line 38: please rephrase, meiosis takes place before gametogenesis during sporogenesis
- line 59: please define abbreviations like AQP upon first usage
- line 383: please refer to database where accession numbers can be found, if applicable
- legend Figure 4: Does “not validated” mean the gene was not significantly differentially expressed in this comparison?
- line 238: why have different photoperiods and temperatures been used in this confirmatory experiment? Please provide information.
- line 314: the statement of similar mapping ratios is not visible in the data, as there is a range between ~50% to 86% mapping
- line 319: sentence starting with "The results indicate..." would benefit from rephrasing
Language related issues
- line 9: better "at three different time points"
- line 11: should it be "numbers of reads" instead of sequences?
- line 44: please rephrase "until cell recovery"
- line 60: polyploidization events and ... duplication of .....
- line 414: growth
- Figure 5: log instead of lgo
Author Response
The manuscript “Comparison of gene expression changes in three wheat varieties with different susceptibility to heat stress using RNA-Seq analysis” targets the important question of transcriptional changes in response to heat stress in wheat. To this aim three wheat varieties are studied, which have previously been shown to differ in resilience and response to heat stress. Greenness index and gene expression was compared after subsequent rounds of heat treatments as compared to non- stressed conditions. Further investigations were focused on genes previously described to play roles in heat stress response. Their expression was further analysed using qPCR.
The topic of the manuscript presented is relevant. However, the analysis of data and the presentation would greatly benefit from thorough revision. Major points and concerns are that the novelty of the results is limited, as authors focused on genes previously described to be responsive to heat stress. Based on the data obtained, in depth analysis would have allowed to gain more comprehensive insights into the changes after heat stress, for example by comparing the sets of differentially expressed genes (DEGs) between the different varieties or by investigation of enrichment of GOs or similar based on the DEGs. Most importantly however, methods used in this study are not presented in sufficient detail. As a minor point, the manuscript would benefit from a careful check for typos etc. For detailed comments please see below.
We appreciate your comments and advice.
GO analysis between the three varieties was performed using DAVID v2022q2 and data were added to the main text. In addition, opinions on materials and methods have been revised and supplemented.
Major comments
1) The data analysis is almost exclusively targeted on groups of genes that were expected to be of interest based on previous knowledge. It would have been interesting to elaborate on the analysis of GO enrichment or protein families or KEGG pathways of DEGs in this context. Furthermore, it would have been of interest to see which DEGs are shared between heat resilient varieties JP and KK and not with OL or if the majority of DEGs are shared.
: GO enrichment analysis data were added to Results section 2.3, Figure 3, Materials and Methods section 4.3.
2) Figure 2: How are non DEGs defined? Are they the genes expressed but not differentially regulated in after stress treatment as compared to the control? Please include an explanation on this.
: Yes. 'non-DEG' refers to a gene that was expressed but not differentially regulated after stress treatment compared to the control group. We have included an explanation of this term in the Figure 2 legend.
3) The presentation of genes, gene models, isoforms and splicing variants is partially confusing. It would be beneficial to provide background information (if possible) on how the genomic background varies between the three varieties. This is relevant for estimation of the preciseness of mapping but also for confirmatory qPCR studies.
: We have added a description of the pedigree of these three varieties to the results section 2.1.
- line 179: Do TraesCS7A02G549800.1 and TraesCS7A02G549800.2 represent different gene models or different splice variants? How precise is the mapping to the distinct variants? Please clarify.
: TraesCS7A02G549800.1 and TraesCS7A02G549800.2 are different splice variants. We have no idea how accurate the mapping of unique strains with RNA-seq analysis between different varieties is.
- line 203f: How was differential splicing of TraesCS1A02G375600.1 addressed? Please clarify.
: Genes for splice variants are available on the website http://plants.ensembl.org. We changed the sentence of ' Two transcripts, TraesCS1A02G375600.1 and TraesCS1A02G375600.1, showed alternatively spliced forms.' to ‘TraesCS1A02G375600 has two transcripts (splice variants), TraesCS1A02G375600.1 and TraesCS1A02G375600.2, (http://plants.ensembl.org.).
4) Methods description is largely lacking sufficient information to follow what the authors have done.
: Added clarifications to the above questions throughout Materials and Methods sections 4.2-4.5.
- line 402: the Illumina NovaSeq 6000 platform is mentioned, but not the sequencing protocol (e.g. read length, single end, paired end), details on potential multiplexing and sequencing adaptors, etc. Please provide detailed informations.
: We've added explanations to your questions above in the Materials and Methods section 4.2.
line 407: EdgeR comprised a variety of different models and methods, please provide all relevant details on the statistics applied, e.g. was it a pairwise comparison, was filtering of low expressed genes applied?
: Added explanation to the Materials and Methods section
- line 408: what was done with the assembler Trinity and how? In other contexts an assembled transcriptome is not mentioned. Please clarify.
: Thanks for pointing it out. As a result of checking the analysis information with the company, it was reported that there was already a reference (Triticum. aestivum) and assembly was not carried out. We corrected this.
- How was counting of mapped reads performed?
: Added explanation to the Materials and Methods section
- For sequencing and also for qPCR please indicate quality and quantity of RNA used.
: Added RNA quality and quantity information to Materials and Methods sections 4.2 and 4.5.
- Please upload raw data to a suitable repository and provide access informations.
: Raw data was included with Additional file 1.1-9, 2.1-3, and 3.1-9.
- How have heatmaps been done?
: Added explanation to the Materials and Methods section 4.5. ‘The heatmap was created in MS Excel using conditional formatting and a color scale using selected genes in Table S3-6.
- Figure 2: padj implies that adjusted p-values have been used. Please provide details.
: Added 'Padj, adjusted p-value' to figure 2 legend.
- line 149: Please provide sufficient details on how cluster analysis was performed.
: Added explanation to the Materials and Methods section 4.4.
- line 219: It is not fully clear what kind of filtering values are meant here? were they removed in filtering of low expressed genes? Or were no significant differences detected? Please clarify.
: These genes were not pass the filtering values CPM (counts per million) 2 after 12 h of heat treatment, such that the corresponding data could not be obtained. Edited the description.
- discussion: For the qPCR results not showing significant differences between stress/no stress the data should not be discussed as confirmation or validation.
: Thanks for the advice. Changed 'validated' to 'detected' and 'investigated' on lines 275 and 369 respectively. Also changed 'validation' to 'Study' on line 384 and changed 'confirmed' to 'investigated' on line 341.
Minor comments:
- line 13ff: context is confusing and should be rephrased. It seems that the numbers provided are average mapping percentages in unstressed control, 6h group, and 12 h group?
: Edited.
- line 38: please rephrase, meiosis takes place before gametogenesis during sporogenesis
: Edited.
- line 59: please define abbreviations like AQP upon first usage
: We defined AQP.
- line 383: please refer to database where accession numbers can be found, if applicableI
: The accession numbers we created is a number used within the institution. So, we modified the number (IT number) and included the address of the site that readers can check.
- legend Figure 4: Does “not validated” mean the gene was not significantly differentially expressed in this comparison?
These genes were not pass the filtering values CPM (counts per million) 2 after 12 h of heat treatment, such that the corresponding data could not be obtained. rephased 'not validated’ to 'not evaluated'.
- line 238: why have different photoperiods and temperatures been used in this confirmatory experiment? Please provide information.
: Corrected for an incorrect 22h photoperiod in plant growth for RNA-seq analysis. Plants for RNA-seq analysis and plants for qRT-PCR experiments were heat-treated at 35°C in the same way. However, in the experiment in which plants were subjected to heat treatment to observe the phenotype, repeated experiments were performed at a high temperature of 45°C to better represent the phenotype caused by heat stress.
- line 314: the statement of similar mapping ratios is not visible in the data, as there is a range between ~50% to 86% mapping
: We tried to express the average value of three replications. However, as pointed out, the mapping rate for each experiment was ~50% to 86% mapping, so we wrote further explain in the manuscripts.
- line 319: sentence starting with "The results indicate..." would benefit from rephrasing
: We changed “These results” to “The results “
Language related issues
- line 9: better "at three different time points"
: Edited
- line 11: should it be "numbers of reads" instead of sequences?
: Edited
- line 44: please rephrase "until cell recovery"
: Edited
- line 60: polyploidization events and ... duplication of .....
: Edited
- line 414: growth
: Edited
- Figure 5: log instead of lgo
: We edited ‘lgo’ to ‘log’ in Figure 3, Figure 4, and Figure 5’.

Reviewer 2 Report
Authors proposed a paper entitled “Comparison of gene expression changes in three wheat varie-2 ties with different susceptibility to heat stress using RNA-seq 3 analysis” for the publication in International Journal of Molecular Sciences, mdpi.
This paper has a good scientific soundeness and deserves to be published after minor revision.
I suggest the addition of an abbreviation list, according to the guidelines of this journal. Just a few examples could be extracted by “peroxidase (APX), catalase (CAT), glutathione reductase (GR), glutathione pe-65 roxidase (GPX), superoxide dismutase (SOD), and peroxidase (POX)”.
I have some few minor issues as follows:
Line 28. “by 4% to 8.5% per degree Celsius” please use just symbols “ %/°C ”
Line 31. A reference is requested here.
Line 36. A double space should be removed here.
Line 37. Problem of format of the paragraph.
The last paragraph of the introductive section should be adapted to the description of the “aims” of the paper. Some descriptions of the modalities and methods should be moved to methods section.
Line 115. “showed many dry leaves”. Do you have a quantitative concentration data?
Please, enlarge figure 1d.
Also improve the quality of figure 2d, the “Jopum” is not completely and clearly visible. Moreover, add standard deviations on these bars, if possible.
Line 270. Are these results comparable with literature?
Line 308. A reference should be added to support this affirmation
In Material and Methods section, Material specific section is missing. Information about raw materials, their manufacturer, manufacturer countries and CAS number should be included.
Author Response
Authors proposed a paper entitled “Comparison of gene expression changes in three wheat varie-2 ties with different susceptibility to heat stress using RNA-seq 3 analysis” for the publication in International Journal of Molecular Sciences, mdpi.
This paper has a good scientific soundeness and deserves to be published after minor revision.
I suggest the addition of an abbreviation list, according to the guidelines of this journal. Just a few examples could be extracted by “peroxidase (APX), catalase (CAT), glutathione reductase (GR), glutathione pe-65 roxidase (GPX), superoxide dismutase (SOD), and peroxidase (POX)”.
: We thank you for the suggestion. We wanted to make a list of abbreviations as you advised, but there are guidelines for displaying abbreviations.
‘Acronyms/Abbreviations/Initialisms should be defined the first time they appear in each of three sections: The abstract, the main text, the first figure or table. When defined for the fist time, the acronym/abbreviation/initialism should be added in parentheses after the writhen-out form.’
So, we reviewed abbreviation in the abstract, the main text, the first figure or table again.
I have some few minor issues as follows:
Line 28. “by 4% to 8.5% per degree Celsius” please use just symbols “ %/°C ”
: We corrected “% per degree Celsius” to “%/°C”.
Line 31. A reference is requested here.
: We added a reference.
Line 36. A double space should be removed here.
: Edited.
Line 37. Problem of format of the paragraph.
: Edited.
The last paragraph of the introductive section should be adapted to the description of the “aims” of the paper. Some descriptions of the modalities and methods should be moved to methods section.
: We edited the last paragraph.
Line 115. “showed many dry leaves”. Do you have a quantitative concentration data?
: We did not quantify dried leaves, only green.
Please, enlarge figure 1d.
: Figure 1d is not included in manuscript. However, we enlarged the size of Figures 1b and 1c.
Also improve the quality of figure 2d, the “Jopum” is not completely and clearly visible. Moreover, add standard deviations on these bars, if possible.
: Figure 2d has been modified to show clearly visible.
In order to answer the advice, information was requested from the analyzed company, but when analyzing DEG with EdgeR, it is said that there is only one expression value for each group by designating the target group for comparative analysis. The company responded that it was difficult to provide again because it was the result of an experiment a long time ago (about 3 years). Sorry for not answering your question.
Line 270. Are these results comparable with literature?
: Do you mean 'the transcriptional levels of CAT-5A and POX-1D in OL were not validated after 9 h of heat treatment (Figure. 7E, 7K)' in line 168-169?
It is reported that the activities of SOD, APX, CAT, GR, and POX were significantly increased at all stages of growth in heat-resistant cultivars in response to heat stress treatment, whereas CAT, GR and POX activities were significantly decreased in susceptible cultivars (Hemantaranjan, A., Bhanu, A. N., Singh, M. N., Yadav, D. K., Patel, P. K., Singh, R., & Katiyar, D. (2014). Heat stress responses and thermotolerance. Adv. Plants Agric. Res, 1(3), 1-10). Although the expression of CAT and POX seems to be lower in Olgeuru, a sensitive cultivar, it is difficult to explain why the expression of CAT-5A and POX-1A was not detected in OL at 9 hours.
Line 308. A reference should be added to support this affirmation
: We added a reference.
In Material and Methods section, Material specific section is missing. Information about raw materials, their manufacturer, manufacturer countries and CAS number should be included.
: Added materials information to the Material and Methods section.

Round 2
Reviewer 1 Report
The revised manuscript “Comparison of gene expression changes in three wheat varieties with different susceptibility to heat stress using RNA-Seq analysis” has been improved by adding new analysis and data (GO analysis and supporting tables). Nevertheless, my concerns and comments originally raised have not fully satisfactorily been addressed. While the presentation of methods has been improved in aspects, there is still information missing that is necessary to present, for example on the statistical approach applied with EdgeR. EdgeR is a statistical package with a variety of distinct moduls and options for data analysis and it is important to clearly state how data analysis has been performed (for example pairwise comparison with classical EdgeR and prefiltering of low expressed reads?).
Furthermore, raw data or a link to raw data uploaded to a suitable repository (for example Dryad or NCBI) are still not presented. Any of the supporting tables provided now contains processed and analysed data, not raw sequencing data.
Also, the presentation of the new data, in particular the GO analysis is not presented in a clear enough manner to enhance the quality of the manuscript. Even though I have been proposing to do a GO analysis or similar the presentation of all three categories of biological process, cellular component and molecular functions for all comparisons in a main figure might not be the most helpful presentation. As presented, even the GO categories at the x-axis are hardly readable. In addition, the description lacks clarity, e.g. what is for example GO 445? What do the authors mean with the statement that CC and MF are similar by giving percentages? Should it mean that similar numbers of DEGs are found in upregulated categories? The meaning of this finding is not fully clear to me and would benefit from a clearer description. Further, it would rather be more helpful to maybe focus on top upregulated categories that are shared under heat stress or differ for varieties and to show only selected findings in the main figure.
Furthermore, the manuscript would still benefit from another round of check for language related issues like typos.
For some additional detailed comments please see below:
- Information would be beneficial in which database accession numbers for varieties can be found.
- line 63f "duplication events in ancestral species"? should it mean events of genome duplication in ancestral species or similar?
- catalog numbers do not need to be provided for consumables
- lines 462ff information is redundant. Please rephrase.
- line 468 what are redundant reads and how were they removed? also the software tool used for filtering and parameters need to be described
- line 471 please mention version of and parameters used for Kallisto
- line 480ff version of EdgeR used, statistical models chosen and parameters used including potential pre-filtering settings need to be given
- line 483 reference to, version number and parameters for Trifiny should be provided. Does it really mean Trifiny or is this a typo?
- lines 134ff which method has been used for p-value adjustment? is the p-value cutoff in line 138 adjusted or non adjusted? Please clarify
Author Response
The revised manuscript “Comparison of gene expression changes in three wheat varieties with different susceptibility to heat stress using RNA-Seq analysis” has been improved by adding new analysis and data (GO analysis and supporting tables). Nevertheless, my concerns and comments originally raised have not fully satisfactorily been addressed. While the presentation of methods has been improved in aspects, there is still information missing that is necessary to present, for example on the statistical approach applied with EdgeR. EdgeR is a statistical package with a variety of distinct moduls and options for data analysis and it is important to clearly state how data analysis has been performed (for example pairwise comparison with classical EdgeR and prefiltering of low expressed reads?).
: Thank you for your kind and detailed explanation of helpful advice. We made general revisions to the Materials and Methods section and added some references. We hope that the information provided in the manuscript's Materials and Methods is sufficient to describe.
Furthermore, raw data or a link to raw data uploaded to a suitable repository (for example Dryad or NCBI) are still not presented. Any of the supporting tables provided now contains processed and analysed data, not raw sequencing data.
: Unfortunately, the former researcher who commissioned the experiment said that he did not store the raw data. So, we asked if the company that does RNA-seq was storing the information, but they said there was no information because the storage period had expired. We are very sorry that we cannot provide raw data.
Also, the presentation of the new data, in particular the GO analysis is not presented in a clear enough manner to enhance the quality of the manuscript. Even though I have been proposing to do a GO analysis or similar the presentation of all three categories of biological process, cellular component and molecular functions for all comparisons in a main figure might not be the most helpful presentation. As presented, even the GO categories at the x-axis are hardly readable. In addition, the description lacks clarity, e.g. what is for example GO 445? What do the authors mean with the statement that CC and MF are similar by giving percentages? Should it mean that similar numbers of DEGs are found in upregulated categories? The meaning of this finding is not fully clear to me and would benefit from a clearer description. Further, it would rather be more helpful to maybe focus on top upregulated categories that are shared under heat stress or differ for varieties and to show only selected findings in the main figure.
:Thanks for the advice. The classification of significantly modulated GO functions with respect to ROS accumulation, cellular response to heat, antioxidant activity, and water channel activity was performed at two different sampling time points after heat treatment of the three cultivars. We included these descriptions in the results section 2.3 and added a figure to the supplementary data (Figure S2).
Furthermore, the manuscript would still benefit from another round of check for language related issues like typos.
For some additional detailed comments please see below:
- Information would be beneficial in which database accession numbers for varieties can be found.
: We included database registration numbers of three wheat varieties.
- line 63f "duplication events in ancestral species"? should it mean events of genome duplication in ancestral species or similar?
: Yes. It means ‘gene duplication events in ancestral species’. Edited.
- catalog numbers do not need to be provided for consumables
: Removed catalog number.
- lines 462ff information is redundant. Please rephrase.
: Removed redundant information.
- line 468 what are redundant reads and how were they removed? also the software tool used for filtering and parameters need to be described
: Additional information included in Materials and Methods section 4.3.
- line 471 please mention version of and parameters used for Kallisto
: Version number and additional information included in Materials and Methods section 4.3.
- line 480ff version of EdgeR used, statistical models chosen and parameters used including potential pre-filtering settings need to be given Include normalization factor,
: Version number and additional information included in Materials and Methods section 4.4.
- line 483 reference to, version number and parameters for Trifiny should be provided. Does it really mean Trifiny or is this a typo?
: Version number and additional information included in Materials and Methods section 4.5.
- lines 134ff which method has been used for p-value adjustment? is the p-value cutoff in line 138 adjusted or non adjusted? Please clarify
: FDR(padj) < 0.05. Edited.
Round 3
Reviewer 1 Report
For the second revised version of the manuscript presented by the authors, some aspects of the manuscript have been improved. But even though the presentation of the GO terms is a bit clearer now, it is still difficult to understand what has been done and what precisely is presented. Have the differentially expressed genes (DEGs) been taken as a basis to perform a GO enrichment study or have DEGs been annotated to the associated GOs without checking for enrichment? I am not familiar with the DAVID tool, however it should be clear also for readers who are not experts for a certain software used.
In line 171, what is meant when saying that “expression of genes” was “not significant”. Are they non-DEGs or is their expression level below a certain threshold that they have been filtered out prior to analysis? What precisely do the percentages given in this section relate to? Is it the percentage of genes annotated in a GO term, which are found in the list of DEGs?
Like already commented in the previous two rounds of review I still believe that the methods description is partially not sufficient, in particular concerning the RNA-Seq data analysis. It cannot clearly be followed, for example, which statistical model EdgeR has been used. Was it a pairwise comparison between conditions that has been performed? Has linear modelling been used? The information should be clarified.
The most critical issue concerning the manuscript is that raw data are not available, which means that the data cannot independently be used, controlled, or confirmed, which is essential for scientific publications. Also, the analyzed data provided for the last submission have now not been included again.
Minor comments
Language check for typos is advised, for example line 480 “grouped”
Line 479ff: information is redundant
Line 467: has the “the soft clustering of time-series gene expression data” really been done using EdgeR? Or was it another software or tool?